# Metal-Filled Polyvinylpyrrolidone Copolymers: Promising Platforms for Creating Sensors

**DOI:** 10.3390/polym15102259

**Published:** 2023-05-10

**Authors:** Oleksandr Grytsenko, Ludmila Dulebova, Emil Spišák, Petro Pukach

**Affiliations:** 1Department of Chemical Technology of Plastics Processing, Lviv Polytechnic National University, 12, St. Bandera Str., 79013 Lviv, Ukraine; oleksandr.m.grytsenko@lpnu.ua; 2Department of Technologies, Materials and Computer Aided Production, Technical University of Košice, 74 Mäsiarska, 04001 Košice, Slovakia; emil.spisak@tuke.sk; 3Institute of Applied Mathematics and Fundamental Sciences, Lviv Polytechnic National University, 12, St. Bandera Str., 79013 Lviv, Ukraine; ppukach@gmail.com

**Keywords:** polyvinylpyrrolidone, 2-hydroxyethylmethacrylate, copolymers, hydrogels, composite hydrogels, metal-filled hydrogels, conductivity, sensors

## Abstract

This paper presents research results on the properties of composite materials based on cross-linked grafted copolymers of 2-hydroxyethylmethacrylate (HEMA) with polyvinylpyrrolidone (PVP) and their hydrogels filled with finely dispersed metal powders (Zn, Co, Cu). Metal-filled pHEMA-gr-PVP copolymers in the dry state were studied for surface hardness and swelling ability, which was characterized by swelling kinetics curves and water content. Copolymers swollen in water to an equilibrium state were studied for hardness, elasticity, and plasticity. The heat resistance of dry composites was evaluated by the Vicat softening temperature. As a result, materials with a wide range of predetermined properties were obtained, including physico-mechanical properties (surface hardness 240 ÷ 330 MPa, hardness number 0.06 ÷ 2.8 MPa, elasticity number 75 ÷ 90%), electrical properties (specific volume resistance 10^2^ ÷ 10^8^ Ω⋅m), thermophysical properties (Vicat heat resistance 87 ÷ 122 °C), and sorption (swelling degree 0.7 ÷ 1.6 g (H_2_O)/g (polymer)) at room temperature. Resistance to the destruction of the polymer matrix was confirmed by the results concerning its behavior in aggressive media such as solutions of alkalis and acids (HCl, H_2_SO_4_, NaOH), as well as some solvents (ethanol, acetone, benzene, toluene). The obtained composites are characterized by electrical conductivity, which can be adjusted within wide limits depending on the nature and content of the metal filler. The specific electrical resistance of metal-filled pHEMA-gr-PVP copolymers is sensitive to changes in moisture (with a moisture increase from 0 to 50%, ρ_V_ decreases from 10^8^ to 10^2^ Ω⋅m), temperature (with a temperature change from 20 °C to 175 °C, ρ_V_ of dry samples decreases by 4.5 times), pH medium (within pH from 2 to 9, the range of ρ_V_ change is from 2 to 170 kΩ⋅m), load (with a change in compressive stress from 0 kPa to 140 kPa, ρ_V_ of swollen composites decreases by 2–4 times), and the presence of low molecular weight substances, which is proven by the example involving ethanol and ammonium hydroxide. The established dependencies of the electrical conductivity of metal-filled pHEMA-gr-PVP copolymers and their hydrogels on various factors, in combination with high strength, elastic properties, sorption capacity, and resistance to aggressive media, suggest the potential for further research as a platform for the manufacture of sensors for various purposes.

## 1. Introduction

Nowadays, in various fields of science and practice, a demand exists for polymer materials that not only possess the necessary physical and chemical properties, but are also able to predictably change their characteristics during their operation depending on external stimuli, due to which they can be used for the manufacture of sensors. Polymer hydrogels are a promising material for creating sensors for various purposes [1,2,3,4]. Polymer hydrogels are characterized by high sorption capacity for low molecular weight substances [5] and permeability to liquids and gases [6], which is a prerequisite for their use in various fields of science and practice [7,8,9]. Such materials have gained a particular practical application in medicine and biotechnology [10,11,12]. Due to their structure, which resembles the structure of living tissues, hydrogels are characterized by biocompatibility, which allows them to be used in direct contact with a living organism [13,14].

Hydrogels represent a particularly interesting class of materials capable of responding to various external stimuli [15]. Polymer hydrogels are loosely cross-linked polymers that can absorb and hold a large amount of liquid [16]. Under the influence of certain external stimuli, such materials change their capacity for water absorption, which in turn affects the change in the volume of the hydrogel [17,18,19]. Due to this property, hydrogel materials are used in sensor devices, in which changes in volume in response to the influence of specific stimuli can be transformed into a measurable signal with the help of a transducer [20].

At present, there is enough information related to research on the use of sensor devices based on polymer hydrogels in various fields. Hydrogel detectors are used to determine chemical compounds, such as ethanol [19,21], glucose [22,23], and ammonia [24,25], and are sensitive to electric fields [26], light [27], temperature changes [17,28], pressure [29], moisture [30,31], and pH [32,33]. However, the use of intelligent hydrogels as sensors is somewhat difficult since a reliable assessment of their swelling state remains problematic. In return, this requires the development of more reliable and highly sensitive methods for detecting the smallest changes in hydrogel volume during its swelling [4]. At the same time, depending on the nature of the polymer matrix of the hydrogel material and the specific stimulus, it can be much more accurate to record changes in other characteristics of the hydrogel. For instance, for electrically conductive hydrogels, this is the effect of the state of the hydrogel on the change in electrical conductivity. However, not all hydrogels are electrically conductive, which limits their ability to respond to external stimuli and transmit signals. Therefore, to provide or increase the electrical conductivity of hydrogels, they must be modified. The most common methods of such modification are the introduction of electrically conductive polymers [34] (such as polypyrrole, polyaniline, and poly (3,4-ethylenedioxythiophene) [35]) or conductive fillers into the structure of hydrogels. Carbon, graphene, and (nano)particles of noble metals are often used as conductive fillers [22,36,37]. Filling polymers with metal particles is one of the methods of obtaining polymer composite materials with specific characteristics [38,39,40,41], in particular electrically conductive ones [42]. The uniqueness of metal-filled hydrogels lies in the fact that they combine the properties of the hydrogel matrix and the metal filler. The introduction of metal particles of various nature into the structure of hydrogels provides them with new properties, including bactericidal and antifungal [43], optical [44], catalytic [45], electrical [46], and magnetic [47] properties, as well as the possibility of changing these properties in the desired direction. Metal-filled hydrogels differ from other metal-filled polymer materials in their ability to change conductive characteristics depending on temperature, pH medium, moisture, and the content of low molecular weight substances, which provides the possibility of their use in the manufacture of various types of detectors and sensors.

The authors of the article developed new composite hydrogel materials based on spatially cross-linked copolymers of polyvinylpyrrolidone (PVP) with 2-hydroxyethylmethacrylate (HEMA) (pHEMA-gr-PVP) filled with finely dispersed particles of metals and alloys of various nature [48]. The growing interest in polymer hydrogels based on polyHEMA has arisen primarily due to the fact that they demonstrate excellent biocompatibility, compatibility with blood, and low thrombogenicity [49,50]. At the same time, hydrogels based on polyHEMA exhibit better mechanical properties compared to hydrogels of other nature [51]. Due to their unique characteristics, hydrogel materials based on polyHEMA are widely used for the manufacture of sensors for various purposes [31,32,33,52]. Polymerization of HEMA in the presence of PVP ensures that pHEMA-gr-PVP copolymers with improved sorption [53], adhesive [54], and elastic [55] characteristics compared to polyHEMA are obtained. At the same time, the great advantage of pHEMA-gr-PVP copolymers is the simple one-step technology of their production in the presence of metal salts of various oxidation states [56]. During the use of PVP/Me^n+^ complex, the copolymerization of HEMA with PVP occurs at a high rate at room temperature, in light, and in the presence of air oxygen whilst also requiring only one stage [57,58]. Depending on the composite’s formulation and the content and nature of the solvent and metal salt, the duration of polymerization of HEMA/PVP compositions (with a maximum polymer yield of 98–99%) can vary within wide limits (from 15 to 150 min). The developed technology combines the stages of polymerization and the formation of hydrogel products, with the possibility of obtaining products of any shape, in particular film [43] and tubular products [59].

The filling of pHEMA-gr-PVP copolymers with metals of various nature provides them with electrically conductive characteristics that require additional research and opens the possibility of using hydrogels based on metal-filled pHEMA-gr-PVP copolymers (Me/pHEMA-gr-PVP) as sensors and detectors. Such materials can possess selectivity and sensitivity in regard to changes in electrical conductivity in response to various stimuli. Therefore, it is of interest to establish the dependence of the electrical properties of composite materials based on Me/pHEMA-gr-PVP copolymers and hydrogels based on them on the formulation of the original composition, the nature of the metal, and the conditions of external influence, including temperature, pressure, moisture, the presence of low molecular weight substances, and pH medium, with the aim of enabling further research and possible use of the developed composites as potential platforms for the manufacture of sensors.

## 2. Materials and Methods

### 2.1. Materials

The following substances were used: 2-hydroxyethylmethacrylate (Sigma Chemical Co, Saint Louis, MO, USA), which was purified and distilled in a vacuum (residual pressure = 130 N/m^2^, T_boil_ = 78 °C); polyvinylpyrrolidone (AppliChem GmbH, Darmstadt, Germany), which was of high purity with MM 12,000 and was dried at 65 °C in a vacuum for 2–3 h before use; and iron (II) sulfate, which was p.a. grade. Finely dispersed metal powders of Zn, Cu, and Co with a particle size of 10 ÷ 50 μm were used as fillers for the experiment.

### 2.2. Synthesis Technique of Me/pHEMA-gr-PVP Composites

To obtain composite Me/pHEMA-gr-PVP copolymers, the polymerization filling method with finely dispersed metal powders was used (Figure 1).

Polymerization of HEMA/PVP compositions filled with metal powders was carried out in the presence of iron (II) sulfate [56,57]. FeSO_4_ was dissolved in 1/3 of the amount of HEMA required for polymerization. In the remaining part of HEMA, the necessary amount of PVP was dissolved and filler was added. The resulting solutions were mixed and stirred during the time τ < τ_v_ (τ_v_ is the viability time of the composition, i.e., the time during which the reaction composition retains fluidity), after which the mixture was dosed into the polymerization mold. Polymerization was carried out at a temperature of T = 25 ± 1 °C for 0.3…1 h (depending on the composite formulation and the nature and amount of metal filler). As a result, metal-filled spatially cross-linked grafted pHEMA-gr-PVP copolymers were obtained [57].

Based on the analysis of previous studies [56,57,60], HEMA in the amount of 70 ÷ 80 mass parts and PVP in the amount of 20 ÷ 30 mass parts were used to obtain composite Me/pHEMA-gr-PVP copolymers. The composite formulation was selected experimentally based on the dependence of the polymerization rate on the content of PVP and FeSO_4_, as well as on viscosity analysis according to the sedimentation conditions of filler particles. Reducing the PVP content in polymer–monomer compositions (PMCs) causes an increase in their fluidity, an increase in viability time (the time when the PMC retains its fluidity), and an increase in the duration of polymerization, which enhances the sedimentation process of the filler. PMCs with higher PVP content are characterized by increased viscosity and are poorly deaerated and dosed. Polymerization was carried out in the presence of FeSO_4_ in the amount of 0.01–0.05 wt.% of PMC mass. In order to combine the stages of synthesis of a hydrophilic polymer and its subsequent swelling in a solvent, the copolymerization of HEMA with PVP in the presence of metal ions can be carried out in the presence of a solvent—water.

### 2.3. Measurements and Characterization

#### 2.3.1. Scanning Electron Microscopy (SEM)

The morphology of the synthesized filled hydrogels in the swollen state was studied under low vacuum conditions using LEO-435VP (Leo Electron Microscopy, Cambridge, UK) apparatus. Swollen samples were cut in water, frozen in liquid nitrogen, and stored at −80 °C for one day before freeze drying. The samples were mounted on double-sided tape on aluminum stubs and sputter-coated with gold/palladium.

#### 2.3.2. Physico-Mechanical Characteristics of Me/pHEMA-gr-PVP Copolymers

The surface hardness (*F*, MPa) of dry samples (with a diameter (Ø) of 12 ± 0.1 mm and height (*h)* equal to 5 ± 0.01 mm) was found according to the conic yield point in a Höppler consistometer (VEB Pruefgeraetewerk, Medingen, Dresden, Germany) at 20 °C by the depth of penetration (*S*, m) of a steel cone with a vertex angle of 58°08′ into a polymeric specimen under a load of *G* = 50 N for 60 s:(1)$$F=\frac{4G}{S^{2}\cdot \pi }\cdot 10^{-6},$$

Boundary water absorption (*W*, %) was determined by the weight method as the difference between dry (*m*_0_, g) and swollen (*m*_1_, g) samples [57]:(2)$$W=\frac{m_{1}-m_{0}}{m_{1}}\cdot 100\%,$$

The kinetics of swelling were characterized by the degree of water absorption α, g (H_2_O)/g (polymer) [57]. α at time t was determined by the ratio of the mass of liquid absorbed by the polymer during swelling (*m_t_*) to the mass of dry polymer (*m*_0_):(3)$$\alpha_{t}=\left(m_{t}-m_{0}\right)/m_{0},$$

Samples 1.3 ± 0.1 mm thick were used for the studies.

Deformation and elastic characteristics of samples in a swollen state, such as hardness number (*H*, MPa), elasticity index (*E*, %), and plasticity index (*P*, %), were determined according to the procedures described in [56] and ASTM D2240-15, the “Standard Test Method for Rubber Property-Durometer Hardness” [61]:(4)$$H=\frac{0.1\cdot G}{\pi \cdot d\cdot h},$$
(5)$$E=\frac{h-h_{1}}{h}\cdot 100\%,$$
(6)$$P={\frac{h}{h}}_{1}\cdot 100\%,$$
where *G* is the applied load, N; d is the diameter of the indenter ball, mm (*d* = 5 mm); *h* is the depth of ball penetration into the sample under load *F*, mm; and *h*_1_ is the residual deformation after removing the load, mm.

#### 2.3.3. Determination of Heat Resistance

Determination of the heat resistance of copolymers was carried out according to the method ISO 306:2013 “Plastics—Thermoplastic materials—Determination of Vicat softening temperature” [62], which consists of determining the temperature at which a standard indenter with a flat bottom surface (Ø 1.128 ± 0.008 mm) is pressed to a depth of 1 mm in the test sample under the action of the load while heated at a constant speed. Studies were performed by using a Höppler consistometer (VEB Pruefgeraetewerk, Medingen, Dresden, Germany).

#### 2.3.4. Study of Chemical Resistance in Aggressive Media

The chemical stability of pHEMA-gr-PVP copolymers was studied using ISO 175:2010 “Plastics—Methods of test for the determination of the effects of immersion in liquid chemicals” [63]. Chemical resistance was determined by the change in the mass of the samples (Δ*M*, %) at a temperature of 23 ± 2 °C after their exposure to different media for 24 and 100 h.

The change in the mass of the sample after each period of study was calculated according to the following formula:(7)$$\Delta M_{i}=\frac{M_{i}-M}{M}\cdot 100,$$
where *M* is the mass of the test sample before its first immersion in the chemical reagent, g; *M_i_* is the mass of the test sample after keeping it in a chemical reagent, g.

#### 2.3.5. Conductivity

The conductivity characteristics of Me/pHEMA-gr-PVP composites were estimated by specific volume resistivity (ρ_V_, Ω m) according to ASTM D4496-21 “Standard Test Method for D-C Resistance or Conductance of Moderately Conductive Materials” [64]. Samples of cylindrical shape with a diameter of (12 ± 0.1) mm and a height of (5 ± 0.1) mm were used for ρ_V_ investigation. The electrical resistance of the samples was studied at a temperature of 20 °C. Measurements of the electrical resistance of the samples in a dry state were carried out after their conditioning for at least 5 h at a temperature of 23 °C and a relative humidity of 50%. Specific volume resistivity was calculated as [48]:(8)$$\rho_{v}=\frac{R_{V}\cdot S}{h},$$
where *R_V_* is volume electrical resistance, Ω; *S* is the square of the sample, m^2^; and *h* is sample thickness, m.

## 3. Results and Discussion

Hydrogel materials suitable for the manufacture of sensors must possess sufficient physico-mechanical and thermophysical properties, sorption capacity, permeability to low molecular weight substances, and chemical resistance in aggressive media in addition to the ability to respond to certain stimuli in many applications. Indicated characteristics are mainly provided by the nature of the polymer matrix of the composite. The object of study in this work is metal-filled hydrogels obtained on the basis of HEMA with PVP copolymers.

### 3.1. Sorption and Physico-Mechanical Properties of Me/pHEMA-gr-PVP Copolymers

Previous studies established that during the polymerization of HEMA/PVP compositions in the presence of Fe^2+^ ions, copolymers are formed, which are characterized by a spatially cross-linked structure formed by blocks of HEMA grafted on PVP [57]. Due to the presence of hydrophilic groups in the structure of the polymer matrix (hydroxyl HEMA and peptide PVP), pHEMA-gr-PVP copolymers [57] and metal-filled composites based on them [48] are characterized by increased sorption capacity for water and other polar solvents. Sorption–diffusion and the characteristics of materials based on pHEMA-gr-PVP copolymers are provided by the developed porous structure of the polymer matrix (Figure 1).

PVP, which is part of the original PMC, as was shown by previous studies [57], does not fully participate in the graft polymerization reaction. After hydration, unreacted PVP is washed out, forming cavities and pores (Figure 1a). The obtained micrographs provide additional information about significant differences in the morphology of swollen hydrogels depending on the presence of metal particles. During polymerization in the mass of the metal-filled HEMA/PVP composition, the formation of a macro-heterogeneous structure is observed (Figure 1b). However, in the case of the composite obtained in the presence of a solvent (Figure 1c), the effect of the filler on the macroheterogeneity of the polymer matrix is not as noticeable as it is during the synthesis of the block copolymer.

The processes of water absorption and swelling were characterized using swelling kinetics curves (Figure 2) and the obtained values of water content (W, %) (Figure 3). It was established that pHEMA-gr-PVP copolymers obtained in the presence of a metal surface belong to materials with limited swelling, which is characteristic for network polymers whose macromolecules are connected by chemical crosslinks. Figure 3 shows the curves of swelling kinetics depending on the formulation of the original composition (a) and metal content (b) for the example of zinc-filled copolymers. The shape of the swelling kinetic curves of such copolymers depends on their composition and the structural parameters of the polymer network. As can be seen from the results (Figure 2a), the rate and degree of swelling naturally increase with an increase in PVP content in the original composition, thus increasing the number of hydrophilic groups in the copolymer structure.

In addition, swelling degree depends on the effectiveness of PVP grafting [57]; ungrafted PVP is washed out during hydration, forming cavities and promoting solvent sorption. A similar regularity is observed for water content (Figure 3a). The introduction of metal powder into the polymer composition causes a change in the degree of intermolecular interaction, which in turn changes the ability of the polymer to swell. As the metal content in the copolymer increases, the swelling rate decreases (Figure 2b). The obtained results can be interpreted as an increase in the number of crosslinks of the spatial network; an increase in metal content causes a decrease in M_C_ [48].

In addition to swelling ability, the obtained metal-filled pHEMA-gr-PVP copolymers have relatively high strength, elastic characteristics, and heat resistance (Figure 3). Synthesized polymers can be operated in two phase states—solid (dry) and elastic (hydrogel in a swollen state)—and properties were studied for dry (Figure 3a) and hydrated (Figure 3b) samples, respectively. Dry samples were studied for surface hardness (F, MPa), water content (W, %), and Vick heat resistance (T_V_), while swollen ones were studied for hardness, elasticity, and plasticity, which were characterized, respectively, by the hardness number (H, MPa), elasticity number (E, %), and plasticity number (P, %).

The introduction of 10 wt.% of metal into the original composition is, in any case, accompanied by an increase in the strength characteristics of filled polymers (both dry and swollen) compared to unfilled. However, the presence of metal particles in the volume of polymers reduces their ability to absorb water and elastic properties and increases plasticity. Strength and elastic characteristics improve with an increase in metal filler content in the composite (Figure 4). At low degrees of filling, a decrease in both the strength characteristics and the water sorption capacity of the composites is observed compared to the unfilled copolymer, which is a consequence of the formation of a defective network. However, as metal content in the original composition increases, the physico-mechanical characteristics increase, while W continues to decrease. Obviously, an increase in strength and elasticity, as well as a decrease in the ability to absorb water, occurs due to an increase in the degree of crosslinking and the number of physical nodes in the polymer network [48].

The effect of the metal is mainly related to adsorption and is associated with the formation of interphase layers, the structure and properties of which are different from the characteristics of the polymer in the volume [48]. At the same time, individual particles of the filler, as a result of physical interaction with the components of the original PMC, can play the role of additional physical crosslinking nodes, compensating for the decrease in the concentration of chemical nodes. The strength characteristics of the copolymer in this case increase almost linearly with increasing filler content.

### 3.2. Chemical Stability of pHEMA-gr-PVP Copolymers

The spatially cross-linked structure of the obtained pHEMA-gr-PVP copolymers provides them with resistance to aggressive media. The study of chemical resistance provides interest in terms of a definition of one of the technical characteristics of the developed materials. The chemical resistance of materials was studied by the change in the mass of the samples (ΔM, %) after their exposure to aggressive media of various nature. The 0.1 N solutions of HCl, H_2_SO_4_, and NaOH, as well as ethanol, acetone, benzene, and toluene, were selected as aggressive media (Figure 5). As can be seen from the obtained results, the mass of material samples in a hydrated state changes slightly in acid and alkali solutions, i.e., the material is chemically resistant to these media. In the acidic medium, mass loss in samples and a change in color from yellow to transparent colorlessness is observed. Color loss indicates the destruction of the complex through charge transfer [57] between HEMA and PVP, and mass loss indicates the release of unreacted physically bound components (HEMA, PVP) that resided in the structure of the complexes. Holding the sample in an alkaline medium causes an increase in its mass, which can be explained by the destruction of physical nodes, which increases the swelling degree of the copolymer. A significant increase in the mass of the material in the C_2_H_5_OH medium occurs as a result of intense hydrophobic–hydrophilic interaction with macromolecules due to hydrogen bonds.

Since materials based on pHEMA-gr-PVP copolymers can be used in an unhydrated state, changes in the mass of unhydrated samples in some solvents (acetone, benzene, toluene) were additionally investigated. It was established that the test material does not adsorb the mentioned solvents and that sample mass decreases in each case, most intensively in the acetone medium. Obviously, this happens due to the displacement of bound water from the sample.

From the conducted studies, it can be concluded that the destruction of pHEMA-gr-PVP copolymers in acidic and alkaline media occurs slowly enough and that materials based on them can be exposed to such media without a noticeable effect on the structure.

### 3.3. Electrical Conductivity of Me/pHEMA-gr-PVP Copolymers

The main characteristic of the obtained metal-filled materials is the appearance of electrical conductivity and its dependence on various factors. The ability of pHEMA-gr-PVP copolymers and metal-filled composites based on them to conduct an electric current was estimated using specific volume electrical resistance (ρ_V_, Ω⋅m). It was established that the electrical conductivity of unfilled copolymers obtained by the block method is typical of dielectrics (Figure 6). The introduction of an electrically conductive filler in the amount of more than 10 wt.% causes the appearance of electrical conductivity in these copolymers, which significantly depends on the nature of the metal filler (Figure 6a).

With an increase in the amount of metal, the specific resistance of the composites decreases intensively up to a filler content of 50 wt.%, after which a slight change in ρ_V_ is observed (Figure 6b). It is important to note that increases in the content of fillers of different nature have a different effect in terms of magnitude on their electrical conductivity (specific resistance). However, a limit exists for the maximum content of each filler in the composition in terms of the point at which the composition still retains fluidity. A further increase in the filler amount makes the composition non-technological, causing difficulties for the removal of air and filling of the forming cavity of the mold.

#### 3.3.1. Effect of Moisture on Electrical Conductivity of Me/pHEMA-gr-PVP Copolymers

Currently, polymer hydrogels are already used for the manufacture of moisture detectors [30]. The introduction of the metal surface to the original composition and the appearance of a new phase in the cross-linked copolymer usually affects the formation of its structure and properties [48]. In particular, the ability to absorb water is affected, the intensity of which can be adjusted within wide limits by changing the composition of the copolymer and the nature of the metal filler. This is especially relevant as it provides the possibility to change the properties of copolymers in a given direction, such as the creation of materials with electrical conductivity that is sensitive to changes in moisture. During a change in the state of the polymer matrix due to swelling or drying of the hydrogel, volume changes and metal–polymer matrix interaction occurs, which affects the distance between the metal-filler particles and, accordingly, the electrical conductivity of the composite. Since Me/pHEMA-gr-PVP copolymers are characterized by the ability to absorb moisture and conduct an electric current, it is important to study the change in their electrical characteristics in a moist environment in order to fully characterize and establish additional possibilities of using such materials. The influence of solvent content during swelling of the hydrogel on the change in its electrical conductivity (electrical resistance) was studied. Polymer samples filled with copper (Figure 7) were used for analysis.

The electrical conductivity of dry samples, which were then immersed in distilled water, was measured. After certain time intervals, their water content and electrical conductivity were determined. In order to identify the influence of the environment on the change in electrical conductivity, tests were carried out at pH = 7 and pH = 2 (Figure 7). As the results show, the change in electrical properties at different pH has the same character during swelling—in each case, there is a decrease in specific volume electrical resistance. However, an intense decrease in resistance takes place in the first stages in conditions of moisture absorption at small amounts of up to 10%. Obviously, this is related to the ionization of pyrrolidone rings, which contribute to the passage of current in the volume. In an acidic medium, the electrical conductivity is somewhat higher, which is a consequence of the sorption by materials of additional electrolyte.

#### 3.3.2. Dependence of Electrical Conductivity of Me/pHEMA-gr-PVP Copolymers on pH Medium

pH medium is a parameter that is of great interest in many fields of use, such as environmental monitoring, biotechnology, and medical diagnostics [32]. Although a sufficient number of pH sensors are currently in use, only a few of them are suitable for the production of disposable detecting elements. The main feature of a disposable detector is a cheap and simple mechanism for converting the necessary information into a signal that can be recorded. In Figure 8, the curves of changes in the specific resistance of metal-filled hydrogels based on Me/pHEMA-gr-PVP copolymers in an acidic medium with different zinc and copper content are presented.

Both for copper and for zinc, the character of the change in specific volume resistance is the same—a sharp decrease in the first minutes of keeping the samples in an acidic medium. The decrease in ρ_V_ is a consequence of the formation of salts in the volume of the hydrogel, which dissociate into ions and cause an increase in electrical conductivity. Comparing the study results of filled hydrogels in acidic and alkaline media, it can be noted that the alkaline medium has a smaller effect on the change in the specific resistance of swollen materials (Figure 9).

Samples filled with zinc are characterized by greater resistance stability during exposure in an alkaline medium compared to an acidic one. The nature of the curves does not change with increasing filler content. It was established that the synthesized copolymers are sensitive to changes in pH in the range of pH = 2–7 (Figure 10), and the change in ρ_V_ resulting from the change in pH significantly depends on the nature and content of the filler.

#### 3.3.3. Effect of the Presence of Low Molecular Weight Substances on the Electrical Conductivity of Me/pHEMA-gr-PVP Copolymers

Sorption capacity and swelling ability provide hydrogels with the possibility of their use as detectors for the presence or determination of the concentration of low molecular weight substances such as ammonia [24] and ethanol [19,21]. Ammonia is an important and key compound in the chemical industry. However, excessively high concentrations of ammonia can be harmful for humans. Therefore, sensors for the detection of ammonia are particularly important for environmental analysis [24]. The development of cheap, efficient, and portable ammonia detectors is of great importance for the protection of the environment and human health [25].

Using the example of ammonia solution (Figure 11a) and ethanol (Figure 11b), the nature of the change in the volume electrical resistance of metal-filled pHEMA-gr-PVP copolymers depending on the residence time in the mentioned environments was studied. For the zinc-filled copolymer, a noticeable change in electrical conductivity is observed during the sorption of low molecular weight substances.

As a result of the sorption of ammonium hydroxide by metal-filled hydrogels, chemical processes occur with the metal in the volume of the sample and complex compounds are formed, the ions of which contribute to the increase in electrical conductivity (a decrease in the specific volume electrical resistance), wherein the most sensitive to NH_4_OH are Zn-filled hydrogels at concentrations [NH_4_OH] > 3 × 10^−4^ mol/l (Figure 11a). As the study results show (Figure 11b), the result of keeping samples of Zn-filled hydrogels in ethanol, i.e., increasing the alcohol content in the hydrogel volume, is an increase in volume resistance. The absorption of ethanol by hydrogels is accompanied by an increase in the volume of samples [48] and, accordingly, an increase in the distance between filler particles, which is the reason for the deterioration of electrical conductivity.

#### 3.3.4. Effect of Load on Changes in the Electrical Conductivity of Hydrogels Based on Me/pHEMA-gr-PVP Copolymers

Soft elastic porous materials are used to make electronic skin and sensors for recognizing and monitoring hand gestures, movement, and human interaction through the use of mechanisms [3]. Due to their good mechanical properties, hydrogels are used not only for the manufacture of strain gauges, but also for pressure detectors [29].

Hydrogels based on Me/pHEMA-gr-PVP copolymers are characterized by high elastic properties [48], which can be used for the determination of changes in their electrically conductive characteristics from applied mechanical force. In this regard, the regularity of the influence of the applied load (compression (P_comp_, kPa)) on the change in the specific resistance of metal-filled hydrogels in the hydrated state was investigated (Figure 12). It was found that with increasing load, ρ_V_ decreases exponentially, while the nature of the metal filler does not affect the behavior of the curve (*ρ_V_ = f*(*P_comp_*)).

Staying in water and swelling provide the composite material with high elastic properties and electrical conductivity. This leads to a more significant change in electrical resistance during loading, which significantly increases the sensitivity of metal-filled hydrogels and their functional properties. It is wise to use such materials for measuring load in wet environments, such as setting concrete, cement, and other construction mixtures, as well as hydrostatic pressure in water.

#### 3.3.5. Effect of Temperature on the Electrical Conductivity of Me/pHEMA-gr-PVP Copolymers

In order to study the effect of temperature on the electrical conductivity of the developed materials, the temperature dependence (ρ_V_) of copolymers filled with Co powder was established (Figure 13a). The obtained data show that the specific resistance of the dry metal-filled copolymer during heating from T = 20 °C to T = 175 °C decreases by 4.5 times, and the sample is smaller after cooling compared to the initial value.

The dependence curve *ρ_V_ = f(T)* passes through a local minimum at a temperature of T = 50 °C and a local maximum at a temperature of T = 100 °C with a further decrease. The decrease in specific resistance at the initial stage is obviously related to the increase in internal stresses in the polymer matrix, which create additional contacts between filler particles. The further growth of ρ_V_ indicates another possible process due to the thermal expansion of the polymer, which is significantly greater than the thermal expansion of the metal particles. The appearance of segmental mobility promotes the distance of the filler particles due to the increase in the thickness of the insulating layer between the particles. The growth of ρ_V_ continues up to the softening temperature of the material (T_V_~100 °C). A sharp decrease in resistivity after 100 °C is possibly related to the tunneling effect [42]. As the theory of electrical contacts shows, the flow of an electric current is possible not only when two conductors are in direct contact, but also when there is an air gap or a dielectric film between them. Tunneling resistance is an exponential function of gap thickness. The resulting dependence (Figure 13b) shows that at temperatures above T_V_, the value of lgρ_V_ is a linear function of the thermal expansion of the polymer. Therefore, an exponential relationship between ρ_V_ and the thermal expansion of the polymer exists. Taking into consideration that the change in the distance between Co particles is directly proportional to the linear expansion of the polymer, then from this assumption it follows that ρ_V_ is an exponential function of the distance between particles. This feature is characteristic for the tunneling mechanism of electron flow through a potential barrier. Thus, it can be assumed that the tunneling mechanism of electrical conductivity is present in the considered system at temperatures T > T_V_. Violation of the linear dependence in T < T_V_ conditions is probably related to conductivity due to the existence of direct contacts between particles, as well as due to the increase in internal stresses in the system.

## 4. Conclusions

New composite materials and hydrogels based on the polymerization of fast hardening HEMA/PVP compositions in the presence of metallic fine fillers (Zn, Co, Cu) were synthesized. The resulting composites are characterized by a unique combination of properties, including the ability to swell in solvents, sorption capacity for low molecular weight substances, high strength and elasticity in the swollen state, and electrical conductivity, which is sensitive to changes in various factors.

It was established that the electrical properties of Me/pHEMA-gr-PVP copolymers are largely determined by their hydrophilicity, as well as the nature and content of the metal filler. The established characteristic feature is the change in the specific resistance of copolymers filled with Cu and Zn depending on the moisture content. In particular, an intense decrease in ρ_V_ occurs during the absorption of moisture in small amounts of up to 10%. The specific resistance of swollen composites is sensitive to a change in the pH medium in the range of pH = 2–7, and the change in ρ_V_ from pH significantly depends on the content and nature of the filler. The sorption capacity of pHEMA-gr-PVP copolymers ensures the absorption of low molecular weight substances, the concentration and nature of which affect the change in the specific volume electrical resistance of the composites, which was studied using the example of C_2_H_5_OH and ammonia solution. Using the high elastic properties of the obtained composites in an equilibrium swollen state, the change in their specific resistance depending on the applied load was investigated; a decrease in ρ_V_ occurs according to an exponential dependence, the nature of which does not depend on the nature of the metal filler. On the example of a Co-filled composite, the effect of temperature on its electrical conductivity (change in electrical resistance) was investigated. It was established that the specific resistance of the metal-filled copolymer during heating from T = 20 °C to T = 175 °C decreases by 4.5 times.

The dependence of electrical conductivity on the nature and content of the solvent, pH, the presence of low molecular weight substances, the applied load, and temperature in combination with a simple production technology ensure the relevance and perspective of studying Me/pHEMA-gr-PVP copolymers in the direction of creating platforms for obtaining sensors for various applications.

## Data Availability

Not applicable.

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
