# Peer review of "Metal-Filled Polyvinylpyrrolidone Copolymers: Promising Platforms for Creating Sensors"

_polymers, 2023, doi:10.3390/polym15102259_

Round 1

Reviewer 1 Report

In this manuscript, it presents a simple one-step technology to synthesis metal-filled pHEMA-gr-PVP(Zn, Co, Cu).TheThe dependence of electrical conductivity on the nature and content of the solvent, pH, the presence of low molecular weight substances, the applied load and temperature in combination with a simple production technology ensure the relevance and perspective of the study of Me/pHEMA-gr-PVP copolymers in the direction of creating platforms for obtaining sensors for various applications. However, there are some unclear descriptions that make it somewhat doubtful. The authors should address several following major issues.

1.     Three metals are mentioned in this paper. Please explain why you make this choice, what is the basis for it, and whether it is related to the formation of metal salt ions.

2.     Figure 2 shows that PVP increases hydrophilic groups, resulting in enhanced the rate and degress of swelling capacity. In Figure 6, 30% wt PVP is selected as the comparison group. Please explain the reason for this choice. (Most of this paper uses 20% wt PVP)

3.     In this paper, it is mentioned that there are many factors affecting the conductivity, but the experimental conditions of this paper are not specified clearly, please add.

4.     In section 3.3.2, Figure 10, in the experiment, the slope of the curve increased with the increase of Zn content at pH 2-7. Please make a reasonable explanation. (No obvious change in Cu)

5.       Relevant literature needs to be referenced and cited. 10.1007/s12613-020-2173-2

Author Response

Dear Reviewer,

Thanks for the review of the manuscript "Metal-filled polyvinylpyrrolidone copolymers: promising platforms for sensors creating" by O. Grytsenko, L. Dulebova, E. Spišák and P. Pukach.

We agree with the comments of the reviewer and take them into account in the work.

Remarks

Answers

1. Three metals are mentioned in this paper. Please explain why you make this choice, what is the basis for it, and whether it is related to the formation of metal salt ions.

Within the scope of this article, the attention was focused on finely dispersed powders of Co, Cu and Zn, as representatives of metals with different chemical activity and different effects on the conductive properties of the obtained composites, namely – Co, as a ferromagnet, Zn – due to its high reactivity towards HEMA/PVP compositions and the sensitivity of electrical conductivity of Zn-filled composites to changes in pH and to the presence of low molecular weight substances; Cu – as a chemically less active filler, but with the best electrical conductivity.

2.     Figure 2 shows that PVP increases hydrophilic groups, resulting in enhanced the rate and degrees of swelling capacity. In Figure 6, 30% wt PVP is selected as the comparison group. Please explain the reason for this choice. (Most of this paper uses 20% wt PVP)

For the preparation of composite Me/pHEMA-gr-PVP copolymers, PVP was used in the amount of 20÷30 mass parts. Such PVP content in the composition is optimal from a technological point of view. The majority of results related to the study of the resistance of swollen samples are presented for the formulation of the original composition HEMA:PVP=80:20 mass parts. However, some comparative results of the study of non-swollen composites (for example, the effect of the nature and filler content on electrical conductivity), were obtained for the PVP content of 30 mass parts.

3. In this paper, it is mentioned that there are many factors affecting the conductivity, but the experimental conditions of this paper are not specified clearly, please add.

In line 213, the conditions for the study of the electrical conductivity of the samples were added: "The electrical resistance of the samples was studied at a temperature of 20ºС. Measurements of the electrical resistance of the samples in a dry state were carried out after their conditioning for at least 5 hours at a temperature of 23ºС and a relative humidity of 50%".

4. In section 3.3.2, Figure 10, in the experiment, the slope of the curve increased with the increase of Zn content at pH 2-7. Please make a reasonable explanation. (No obvious change in Cu).

It is obvious that the slope of the curve ρv=f(pH) in the range of pH 2-7 with an increase in the content of Zn and Cu in the hydrogel is affected by the chemical activity of the metal-filler.

Zinc reacts with many acids (in particular HCl, which was used in the work for creation the appropriate pH). Therefore, an increase in the Zn content leads to an increase in the content of additional ions in the hydrogel, which cause a decrease in electrical resistance (an increase in electrical conductivity).

Copper is a chemically less active metal and does not interact with hydrochloric acid. Therefore, the resistance of hydrogels is affected only by an increase in the content of conductive filler.

This is a valid observation. However, one of the tasks of the presented results in the article was to show the ability of synthesized composites to respond to various factors by changing their electrical conductivity. In further works, more detailed studies of the obtained materials are planned, in particular, as sensors for specific applications, including as pH detectors with a thorough explanation of the occurring processes.

5. Relevant literature needs to be referenced and cited. 10.1007/s12613-020-2173-2

Based on the reviewer's recommendation, the literature was cited https://doi.org/10.1007/s12613-020-2173-2 (position [44], line 639):

Yang, T.; Zheng, Yp.; Chou, KC. Hou, Xm. Tunable fabrication of single-crystalline CsPbI3 nanobelts and their application as photodetectors. Int. J. Miner. Metall Mater. 2021, 28, 1030–1037. https://doi.org/10.1007/s12613-020-2173-2

Reviewer 2 Report

The manuscript by  Oleksandr Grytsenko et al. ,entitled “ Metal-filled polyvinylpyrrolidone copolymers: promising platforms for sensors creating” submitted to polymers, study a composite based on  cross-linked grafted copolymers of 2-hydroxyethylmethacrylate (HEMA) with polyvinylpyrroli-13 done (PVP) and their hydrogels filled with finely dispersed metal powders (Zn, Co, Cu). This kind of composite has been studied and the results published by the authors( refs. 48, 57, etc .in the manuscript). The authors repeat in this manuscript many information already published which must be avoided or added to Supporting Information (SI).

The interest of the manuscript reside in the plication of theses composites for sensing and so the introduction should be focused on this topic and some  experimental details already published moved to SI.

The possibility of using these composites as sensors based on their  electric conductivity is interesting and must be further explored, namely as pressure and pH sensors. Interesting, is also  the possibility of using these composites to probe small molecules. In this case a more detailed study  must include  selectivity and interference studies, the time response and the limit of detection.  

Author Response

Answers on Reviewer’s 2 remarks

Dear Reviewer,

Thanks for the review of the manuscript "Metal-filled polyvinylpyrrolidone copolymers: promising platforms for sensors creating" by O. Grytsenko, L. Dulebova, E. Spišák and P. Pukach.

We agree with the comments of the reviewer and take them into account in the work.

Remarks

Answers

Moderate English changes required

The presentation of the material in English has been corrected in the following lines:

83 – (gives them) provides them with

213 – (study) investigation

240 – (enter into) participate in

500 – (in) of

523 – This section is not mandatory but can be added to the manuscript if the discussion is unusually long or complex.

This kind of composite has been studied and the results published by the authors ( refs. 48, 57, etc .in the manuscript). The authors repeat in this manuscript many information already published which must be avoided or added to Supporting Information (SI).

The peer-reviewed article is a logical continuation of the previous works of the authors – [57] and [48]. All publications are united by a single common goal – the synthesis and research of HEMA with PVP copolymers and metal-filled composite materials based on them. At the same time, the tasks set in each publication are different, without exception in the peer-reviewed article. Therefore, for the qualitative assessment of the results of the peer-reviewed work, we use references to previous publications. However, for a better perception and understanding of the obtained results, as well as for comparative purposes, in the graphic part there are elements of images and results (which are also referenced), which we believe ensure the integrity of the obtained results within the presented purpose and objectives of the publication.

For example, Fig. 1a, which is presented in [57], is necessary for analyzing the effect of the presence of metal filler particles on the formation of the structure of copolymers. The presence of the results of the CO effect on the physico-mechanical characteristics of composites (Fig. 3) and electrical resistance (Fig. 6) are necessary in this article for a comparative assessment of the influence of the metal-filler nature on the stated characteristics.

We agree that it is appropriate to remove some elements from the peer-reviewed paper (Scheme 2, line 228), and leave only the reference made in line 227.

All other results described in the peer-reviewed article are not presented in publications [57] and [48].

At the same time, we are ready to consider the reviewer's specific suggestions for improving the quality of the article.

The interest of the manuscript reside in the plication of theses composites for sensing and so the introduction should be focused on this topic and some  experimental details already published moved to SI.

In the introduction, the authors aimed to show the relevance of using metal-filled hydrogels as sensors for various purposes, the electrical conductivity of which is sensitive to changes to various external factors, in particular, the prospect of using metal-filled hydrogels based on HEMA with PVP copolymers, as well as the need to study their conductive characteristics depending on the influence of stimuli of different nature. In our opinion, the introduction is informative, highlights scientific research of recent years, including the authors of the manuscript, and fully reflects all the allotted tasks.

The possibility of using these composites as sensors based on their electric conductivity is interesting and must be further explored, namely as pressure and pH sensors. Interesting, is also  the possibility of using these composites to probe small molecules. In this case a more detailed study must include  selectivity and interference studies, the time response and the limit of detection.  

At this stage of research, we have established the possibility and proposed a simple technology for obtaining metal-filled HEMA with PVP copolymers, which are able to respond by changing electrical conductivity to external stimuli of various nature. The goal is to show a wide range of possibilities of the obtained composites as potential materials for the manufacture of sensors for various purposes. Further studies will present more detailed studies of the obtained materials as sensors for specific applications, including as pressure and pH detectors.

Round 2

Reviewer 1 Report

Accept!

Reviewer 2 Report

The answers of the authors, although not addressing all my comments, are enough to consider appropriate the publication of the manuscript in the present form.